# CausalFusion: Integrating LLMs and Graph Falsification for Causal Discovery

## Abstract

Causal discovery is central to enable causal models for tasks such as effect estimation, counterfactual reasoning, and root cause attribution. Yet existing approaches face trade-offs: purely statistical methods (e.g., PC, LiNGAM) often return structures that overlook domain knowledge, while expert-designed DAGs are difficult to scale and time-consuming to construct. We propose CausalFusion, a hybrid framework that combines graph falsification tests with large language models (LLMs) acting as domain-specialized data scientists. LLMs incorporate domain expertise into candidate structures, while graph falsification tests iteratively refine DAGs to balance statistical validity with expert plausibility. We evaluate CausalFusion through two experiments: (i) a synthetic e-commerce dataset with a precisely defined ground truth DAG, and (ii) real-world supply chain data from Amazon, where the ground truth was constructed with domain experts. To benchmark performance, we compare against classical causal discovery algorithms (PC, LiNGAM) as well as LLM-only baselines that generate DAGs without iterative falsification. Structural Hamming Distance (SHD) is used as the primary evaluation metric to quantify similarity between generated and "true" DAGs. We also analyze different foundational models chain-of-thought traces to examine whether deeper reasoning correlates with improved structural accuracy or reproducibility. Results show that CausalFusion produces DAGs more closely aligned with ground truth than both classical algorithms and LLM-only baselines, while offering interpretable reasoning at each iteration, though challenges in reproducibility and generalizability remain.

## 1 Introduction

In the intricate web of modern businesses, identifying the true drivers of critical outcomes is a formidable challenge, whether it is a failed product launch, a sudden spike in employee attrition, or rising delivery delays. Typical methods for root cause attribution rely heavily on descriptive analytics, correlations, or hierarchical "waterfall" attribution logic. Waterfall attribution, which is widely deployed across large organizations, assigns responsibility through a predefined sequence of checks and rules. Its appeal lies in its simplicity and ease of communication for decision-making. Yet it has important limitations: it relies on heuristics (rule-based logic that is not directly grounded in observed data) and is not causal by design, since it neither accounts for confounding nor identifies true causal pathways. SHAP values, for instance, can attribute an outcome to individual features by distributing contributions among them (Lundberg and Lee (2017)), which helps compensate for the heuristic limitation, but these methods remain fundamentally associational and do not address the causal gap.

Double machine learning (DML) (Chernozhukov et al. (2018)) is well-suited for robust estimation of causal effects in semi-parametric settings, addressing both gaps. DML, however, is focused on quantifying causal effect of a treatment on an outcome, given a number of covariates. Another important causal framework is that of Structural Causal Models (SCMs). SCMs are non-parametric and therefore exhibit slower convergence to the ground truth, but they extend well beyond effect estimation to support richer tasks such as mediation analysis, counterfactual ("what-if") reasoning, and root cause attribution (e.g., identifying and ranking the main contributors to an outcome). This breadth of capabilities is associated with more assumptions, however it makes SCMs particularly suitable for root cause attribution in business contexts (if assumptions hold), where the objective is

to disentangle the contributions of multiple potential drivers rather than focus narrowly on a single treatment–outcome relationship. The flexibility (and assumption heaviness) of SCMs derives from their backbone: the causal Directed Acyclic Graph (DAG). Unlike associative models, which capture correlations without specifying directionality, causal DAGs explicitly encode causal pathways and represent all variables as first-class entities in the system. This contrasts with DML that prioritizes a single treatment and outcome while relegating other variables to the role of nuisance covariates used only for adjustment.

A key challenge in scaling SCMs lies in the construction of the underlying causal DAG. Purely data-driven discovery algorithms, such as the PC algorithm (Spirtes et al. (2000)), rely on conditional independence patterns. Other approaches, such as LiNGAM (Shimizu et al. (2006)), infer edge directions under stronger assumptions of linear functional relationships and non-Gaussian noise. While effective in some settings, both these and other existing methods known to the authors hinge entirely on statistical relationship between the variables and are agnostic to domain knowledge unless manually included. Manual DAGs, on the other hand, incorporate expert understanding but remain inherently subjective and scarcely scalable.

In addition to purely data-driven approaches, several causal discovery frameworks allow researchers to inject domain knowledge to guide the search for causal structures. Constraint-based methods such as the PC and FCI algorithms can incorporate background knowledge in the form of forbidden or required edges, or temporal orderings, ensuring that the output DAG respects expert-specified constraints. Score-based methods such as GES and GIES similarly allow priors over edges, either as hard restrictions or as probabilistic weights that bias the scoring function (Hauser and Bühlmann (2014)). Continuous optimization approaches, such as NOTEARS, also support knowledge injection via masks that forbid or enforce specific edges (Zheng et al. (2018)). Finally, hybrid frameworks like TETRAD, bnlearn, and DoWhy provide user-friendly interfaces for integrating expert DAG fragments, structural priors, or whitelists/blacklists into the discovery process (Ramsey et al. (2018)).

In purely data-driven causal discovery, many DAGs remain indistinguishable because they belong to the same Markov equivalence class (MEC), limiting identifiability from observational data alone. Injecting domain knowledge has therefore become a strategy to reduce ambiguity. Constraint-based approaches such as PC and FCI show that even a small number of forbidden or required edges can eliminate large regions of the search space. Score-based methods such as GES and GIES extend this principle by encoding priors as penalties or probabilistic weights in the scoring function, formalizing expert beliefs in a quantitative way (Hauser and Bühlmann (2014)). Continuous optimization methods like NOTEARS demonstrate that knowledge injection can also be operationalized as structural masks that forbid or enforce edges in differentiable programs (Zheng et al. (2018)). Hybrid frameworks such as TETRAD, bnlearn, and DoWhy provide practical interfaces for integrating expert DAG fragments, structural priors, or whitelists/blacklists (Ramsey et al. (2018)). Although these methods enable domain knowledge to be incorporated into data-driven DAG generation, they still rely on researchers or domain experts to undertake the subjective effort of translating general expertise into formal structural assumptions. CausalFusion seeks to automate this step.

Some recent works explore LLM-based approaches to causal discovery have been explored (Vashishtha et al. (2023); Mullapudi et al. (2025)). These methods rely entirely on the domain knowledge embedded in the LLM's training data and therefore operate primarily on variables metadata, like, variable names and descriptions, rather than observed data. By contrast, the present work proposes framework that combines "traditional" methods, rigorously grounded in statistical relationships, with LLM-based ones, which are leveraged to integrate domain knowledge into the discovery process.

Merging these two approaches also represent the integration of two philosophical perspectives on causality. LLM-based approaches align with a subjectivist perspective, in which causal relationships are defined by the knowledge embedded in the model's training data, reflecting the accumulated understanding and consensus of a community, but also carrying its biases and limitations. In turn, "traditional" methods align with an objectivist perspective where real entities in the world are identified with their observed measurements, and causal relationships are derived from the statistical dependencies between such measurements. Each view has its shortcomings: subjectivist knowledge may be incomplete or inaccurately documented, while objectivist inference may fail in cases of noisy data or weak statistical signals. By combining the two, a hybrid approach has the poten-

tial to produce more robust results, compensating for the absence or unreliability of either form of evidence.

We therefore propose CausalFusion, a causal discovery framework in which the LLM acts as a domain-specialized data scientist agent tasked with constructing a causal DAG. The agent first generates candidate structures informed by domain knowledge and then, within a feedback loop, evaluates their consistency with observed data through conditional independence tests. Finally, it proposes one or more DAGs that balance expert knowledge and statistical validity, while motivating the trade-offs between the two. This feedback loop provides a scalable foundation for subsequent SCM-based causal tasks, such as root cause attribution.

## 2 METHODOLOGY

The proposed framework integrates foundational model(s) with a knowledge base of causality and domain expertise. It operates in an iterative loop beginning with the construction of an initial DAG informed by domain knowledge. This candidate structure is then subjected to falsification procedures, which evaluate the extent to which the proposed DAG is consistent with the observed data. Both the DAG and the associated validation results are then returned to the LLM, which refines the graph, provides the thought process behind the refinement procedure and assigns a confidence score to the updated structure.

The iterative process terminates when either the assigned confidence score exceeds a predefined threshold $\gamma$ or the maximum number of iterations $\alpha$ is reached. At convergence, the framework outputs the top $\beta$ DAGs with the highest confidence scores. Multiple output DAGs are allowed because (i) falsification tests often admit several structures that fit the data equally well (i.e., belong to the same Markov equivalence class), and (ii) multiple candidate graphs may be consistent with the specified domain knowledge. The following sections detail the components of the framework, highlighting how each contributes to addressing the challenges of causal discovery in complex domains.

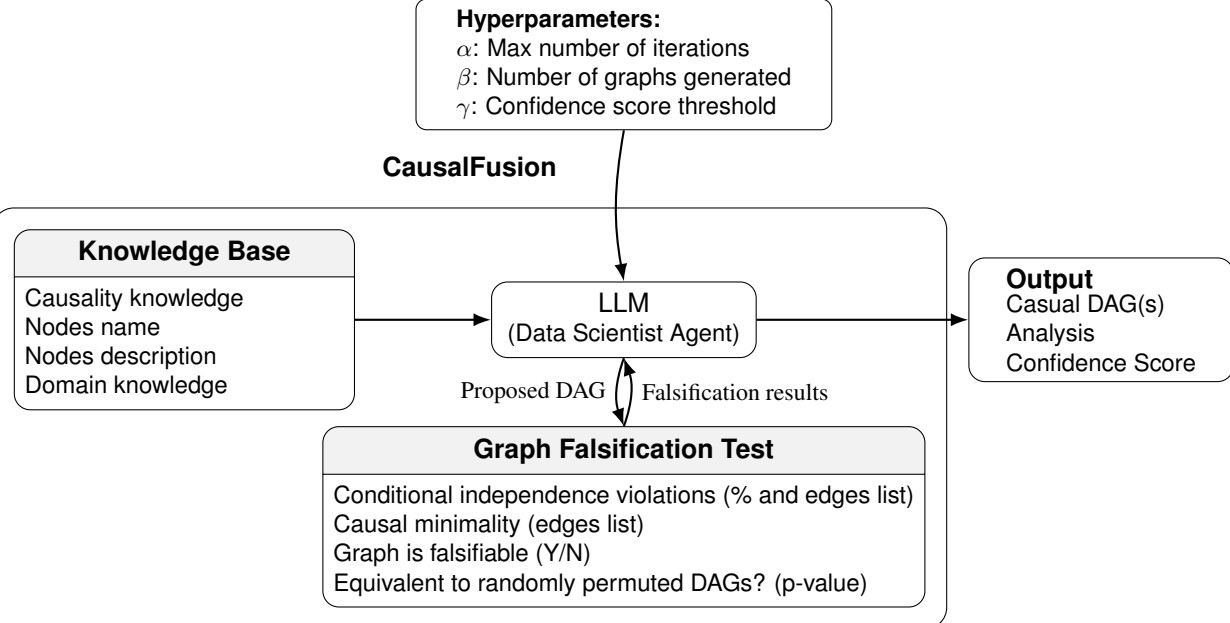

Figure 1: CausalFusion design

### 2.1 KNOWLEDGE BASE

The LLM component is designed to operate with a knowledge base that integrates principles of causality with domain expertise (if not already captured in the pretraining corpus, see Section 4.1.1). This design choice reflects the methodological need to ground LLM reasoning in both theoretical and applied foundations, allowing the LLM to act as a data scientist agent specialized in a given

domain. While the domain-specific knowledge may vary depending on the application, the causality principles are domain-agnostic and provide the scaffolding required to interpret falsification tests and refine DAGs. The causality knowldege is structured as follow.

- **Statistical foundations.** The description and formal definition of the falsification tests provide the essential basis for the LLM to interpret their outcomes (see Section 2.3).

- **Software documentation.** To provide an in-depth understanding of the falsification tests and their practical properties, we include the official `DoWhy` documentation for the `falsify_graph` function (PyWhy Community (2023a)). This enables the LLM to bridge the gap between the theoretical definitions of the tests and their real-world implementation. In particular, it makes the LLM aware of considerations such as implied significance thresholds and the method selected to test conditional independencies (e.g., kernel-based methods, linear correlation tests, etc.). Such details are needed for assessing the validity of results in practice, especially when combined with domain-specific knowledge. For instance, the LLM may notice if the leveraged data types are poorly suited to the employed methods, and therefore disregard falsification results in favor of domain-knowledge.

- **Causal expert knowledge.** To enhance the agent's capability to act as an expert data scientist, we include critical insights and best practices obtained through forums within causality research communities (e.g. PyWhy Community (2023b)), as well as in-person discussions with peers. This knowledge was formalized to capture the practical limitations of falsification tests in real-world settings and to distill best practices, enabling the LLM to better navigate the trade-offs between domain knowledge and empirical validation. A summary of such limitations included in the knowledge base is provided below:

  - Falsification is one-sided: the test can only reject a DAG but not confirm it's the true one.
  - Inability to resolve Markov equivalence: many DAGs imply the same set of conditional independence constraints.
  - Vulnerability to unobserved confounding: if relevant confounders are missing from the data, the test may validate an incorrect DAG.
  - Dependence on measurement quality: the test assumes that observed variables accurately represent the true causal variables. Noisy, aggregated, or proxy measurements may distort the independence structure, leading to spurious validation or falsification.
  - Benchmarking against random graphs may be misleading: even when a graph outperforms random permutations (e.g., $p < 0.05$), the absolute number of independence violations may still be high. In such cases, the graph cannot be considered reliable in practice despite passing the statistical benchmark.
  - Limitations of causal minimality checks: removing edges increases the number of implied CIs. Accordingly, removing unnecessary edges can lead to more CI violations.

## 2.2 LLM COMPONENT

The LLM component of CausalFusion serves as the reasoning engine that integrates statistical evidence with domain knowledge to iteratively refine candidate DAGs. At each iteration, the LLM receives a prompt consisting of: (i) the agent role specification (expert data scientist), (ii) the current DAG (from a prior iteration or, in the first iteration, generated from base knowledge), (iii) the falsification results of the current DAG, and (iv) explicit knowledge boundaries. To ensure consistency and parseability, the prompt further incorporates few-shot demonstrations of the desired output format, a structured task description (analyze falsification results, propose a refined DAG, and assign a confidence score), and a standardized schema to facilitate downstream extraction. In contrast to existing approaches, our framework generates richer insights: (i) a refined DAG, (ii) a rationale that explains the refinement choices, and (iii) a confidence score reflecting the model's self-assessment of how closely the output DAG would align with the ground truth. These outputs may be tied to the depth of the reasoning process performed by the foundational models.

We therefore hypothesize that models with greater chain-of-thought (CoT) capacity (i.e., the ability to generate longer, more structured reasoning traces) will produce DAGs more closely aligned with ground truth. In particular, causal graph construction is inherently a multi-step reasoning task,

requiring the integration of metadata (variable names and descriptions), domain-specific causal assumptions, and falsification evidence. An excerpt of the agent interaction log is shown in *Appendix A*.

## 2.3 GRAPH FALSIFICATION COMPONENT

In CausalFusion, graph falsification tests are reinterpreted as structured feedback to guide the iterative refinement process driven by the LLM component. Each falsification output is surfaced as an interpretable signal that the LLM can incorporate when proposing adjustments to the candidate DAG. This design repurposes DoWhy falsification framework (Sharma and Kiciman (2020); Blöbaum et al. (2022)) to act as a communication channel between statistical testing and natural-language reasoning, thereby enabling a hybrid workflow that neither component could achieve on its own. We next formalize the set of tests employed in this framework and their role in evaluating candidate causal graphs. We include these tests because together they cover the two main sources of structural error: unnecessary edges (causal minimality) and missing edges (CI violations). This provides both local diagnostics, such as lists of problematic edges, and global metrics assessing the DAG as a whole.

### 2.3.1 CONDITIONAL INDEPENDENCE TESTS

To provide the LLM with evidence about whether the implied dependencies of a DAG are consistent with the observed data, we first test CI relations implied by the Local Markov Condition (LMC). LMC asserts that each variable is conditionally independent of its non-descendants given its direct parents. Formally, for a variable $X_i$ in a DAG $\mathcal{G}$, this condition implies: $X_i \perp\!\!\!\perp \text{NonDescendants}(X_i) \mid \text{Pa}(X_i)$.

Under this assumption, the DAG encodes a set of CIs that must hold in the observational distributions if the DAG is valid. CIs are evaluated through hypothesis tests. We selected a kernel-based method to perform such tests due to its flexible, non-parametric nature, making it well-suited for capturing non-linear dependencies and handling mixed data types, as present in our dataset. If an implied CI statement is rejected at the 0.05 significance level, the CI is considered falsified. To avoid a purely binary interpretation by the LLM, the $p$-value is also provided alongside each CI test. Finally, the fraction of violated CIs over the total tested is reported as a global validity metric, while the results of individual CI tests are returned as localized diagnostics.

### 2.3.2 CAUSAL MINIMALITY

To inform the LLM about whether edges in the DAG are necessary, we next test for violations of causal minimality. A causal DAG is said to be *causally minimal* if every edge in the graph corresponds to a necessary causal relationship. In other words, for every edge $X \to Y$ in the graph $\mathcal{G}$, the variable $Y$ is not independent of $X$ given its other parents: $Y \not\perp\!\!\!\perp X \mid \text{Pa}(Y) \setminus \{X\}$.

This condition ensures that each parent variable contributes unique information to its child that cannot be explained away by the remaining parents. Removing such an edge would eliminate a statistical dependency supported by the data, making the graph incomplete. While causal minimality is not a required assumption for defining a causal DAG, it is often imposed in practice to avoid overfitting the graph to spurious associations at a later stage (e.g. when fitting a SCM). For every node $Y$ and each of its parents $X \in \text{Pa}(Y)$, we test if the edge is causally minimal with the same kernel-based method used for CI tests. If $Y$ is independent of $X$, $X \to Y$ is considered unnecessary. Finally, a list of all unnecessary edges is returned. Note that removing edges from the graph increases the number of implied CIs. Consequently, reducing seemingly unnecessary edges can impact the fraction of CI violations, making the process of optimally adjusting the DAG structure inherently non-trivial for the LLM.

### 2.3.3 FALSIFIABILITY AND RANDOM GRAPH BENCHMARK TEST

Before the random graph benchmark test is applied, the LLM must know whether its results are meaningful; this is established by a falsifiability check. Falsifiability is assessed using Markov equivalence classes (MECs), which group DAGs that imply the same CIs and are therefore indistinguishable from an associational perspective. The algorithm measures how often random node

permutations of the graph fall into the same MEC as the candidate. If more than 5% of permutations are in the same MEC, the DAG is deemed not falsifiable as a significant number of possible DAG configurations are statistically indistinguishable. Alongside the falsifiability check, the LLM also receives the results of benchmark against random graphs. This test evaluates whether the candidate DAG performs significantly better than randomly permuted alternatives. Specifically, node permutations are generated, and the number of CI violations associated to the permuted DAG is compared to that of the candidate DAG. If more than 5% of the permuted DAGs exhibit fewer violations, the candidate DAG is deemed falsified.

## 3  EXPERIMENTAL DESIGN AND RESULTS

The experimental design of this study is structured around three research questions (RQs) concerning the CausalFusion framework:

- **RQ1 - Accuracy.** How does CausalFusion perform compared to established causal discovery algorithms, specifically PC and LiNGAM, as well as purely LLM-based baselines where a DAG is generated directly from prompting without iterative falsification?

- **RQ2 - Reproducibility.** Are the results reproducible across repeated runs under identical conditions? Since stability of the learned DAGs is an ideal feature to ensure trustworthiness in both scientific and applied settings, reproducibility is evaluated.

- **RQ3 - Reasoning depth.** Do different foundation models exhibit systematic differences in reasoning depth, as measured by the number of tokens generated when the LLM produces explanations for its proposed DAG? If so, do deeper reasoning traces correlate with (i) improved accuracy in DAG generation (RQ1) or (ii) increased reproducibility across runs (RQ2)?

In the following sections, we detail the experimental design corresponding to each research question. All experiments are conducted with three foundational models—Claude 3.7 Sonnet, Nova Pro, and Nova Premier, selected for their assumed differences in chain-of-thought (CoT) capability. This design allows us to assess how variation in the underlying model influences the framework, an aspect not explored in prior work on employing LLMs for causal discovery.

All experiments were conducted with hyperparameters $\alpha = 5$, $\beta = 1$, and $\gamma = 0.80$. These values were chosen heuristically to balance tractability and stability: $\alpha = 5$ allows for a modest number of iterations, $\beta = 1$ isolates the effect of a single candidate graph per run, and $\gamma = 0.80$ provides a reasonable confidence cutoff.

### 3.1  RQ1 - ACCURACY

Classical benchmarks such as Sachs Sachs et al. (2005), Alarm Beinlich et al. (1989), and Asia Lauritzen and Spiegelhalter (1988) offer established ground-truth DAGs for evaluating causal discovery methods. However, their widespread use makes them likely candidates for inclusion in the pretraining data of modern foundation models, introducing a risk of knowledge leakage when applied to LLM-based evaluation. To address this, we design two complementary experiments. The first uses a synthetic dataset with a precisely defined ground truth DAG, allowing controlled evaluation of structural accuracy under theoretical conditions. The second employs real-world data from Amazon's supply chain operations, where a ground truth DAG was constructed with domain experts. This setting introduces noisy measurements and company-specific terminology outside the LLM's pretraining, offering a more challenging and realistic test. Together, these experiments aim to balance internal validity (synthetic, controlled) and external validity (real-world, applied).

### 3.1.1  RQ1: EXPERIMENTS

The first experiment focuses on the Capped Out Hours (COH) metric in Amazon's logistics operations. COH occurs when a delivery station operates at or beyond its maximum processing capacity, such that no additional packages can be handled beyond those already scheduled. In practice, this is measured as the number of hours a station remains in this "maxed out" state, reflecting periods of severe congestion. As a critical operational KPI, COH is influenced by multiple upstream factors, including demand forecasts, inbound volume, capacity planning, and backlog dynamics. To establish

a credible ground truth DAG, we collaborated with domain experts in supply chain operations who systematically defined the causal structure based on their knowledge of system dynamics. Node descriptions and their downstream effects are detailed in *Appendix B*. This expert-informed DAG serves as the benchmark against which we assess the quality of graphs generated by CausalFusion.

The second experiment leverages a synthetic dataset designed to emulate causal structures commonly found in e-commerce. Following the standard practice of evaluating causal discovery methods on simulated graphs (Heinze-Deml et al., 2018), we construct a DAG inspired by typical business processes (e.g., advertising → website traffic → conversion rate → revenue). Importantly, while the structure is inspired by canonical relationships, it is not directly borrowed from existing benchmarks. This deliberate modification helps ensure that the ground truth DAG is unlikely to appear in the training corpus of foundation models, reducing risks of data leakage. The dataset is generated by sampling from structural equations with both linear and nonlinear functional forms, reflecting the diversity of relationships in real-world business settings. For instance, website traffic depends linearly on advertising spend and seasonal effects, while conversion rate includes nonlinear saturation effects through clipping and interaction with customer satisfaction. Noise terms are added as independent Gaussian, exponential, beta, and uniform components, ensuring stochasticity and heterogeneity across nodes. No additional domain knowledge was supplied in this experiment, as the domain is general and does not involve company-specific causal structure or terminology, allowing us to assume that the LLM already possesses the necessary background knowledge. The ground truths for both experiments are presented in *Appendix C* and *Appendix D*, respectively.

### 3.1.2 RQ1: RESULTS

To measure the similarity between generated DAGs and the ground truth, we adopt the structural Hamming distance (SHD). SHD is defined as the minimum number of edge additions, deletions, or reversals required to transform a candidate DAG into the ground truth. Lower SHD values indicate closer alignment with the true causal structure. Unlike local edge-based metrics such as precision, recall, or $F_1$, which emphasize edge-wise correctness, SHD captures global structural differences between graphs. Indeed, two DAGs with similar precision and recall scores may nevertheless differ substantially in topology, leading to divergent causal implications under interventions. We acknowledge that SHD evaluates structural similarity rather than causal effect estimation. Although interventional differences (such as comparing the distributions implied by estimated graphs and the ground truth under interventions) offer a complementary perspective, their evaluation lies beyond the scope of this study and is left for future work. Our focus here is explicitly on causal discovery, i.e., the recovery of the correct graph structure.

The results in Table 1 highlight clear differences in performance between traditional causal discovery methods and the proposed CausalFusion framework. On the COH dataset, both PC and LiNGAM produced substantially higher SHD values (25 and 22, respectively), indicating poor alignment with the domain-expert ground truth. By contrast, CausalFusion consistently recovered graphs closer to the reference, with Nova Pro achieving the lowest SHD (3) at a confidence level of 0.80. On the synthetic retail dataset, a similar trend is observed: CausalFusion outperforms PC and LiNGAM, with Nova Pro again producing the closest approximation to the ground truth (SHD = 2). Taken together, these findings suggest that CausalFusion provides more accurate causal structures than existing methods in both applied and theoretical settings, and that performance varies across foundational models. Interestingly, Nova Premier, while reporting the highest confidence score (1.00) in both experiments, did not achieve the lowest SHD. Examination of its analysis outputs shows that Nova Premier prioritized improving falsification outcomes far more strongly than the other two models under the same prompt. This behavior resembles the constraint-based logic of the PC algorithm, which relies heavily on conditional independence testing, and might explain the similarity in their results. It may also help explain why Nova Premier performs worse within CausalFusion, where falsification tests are iteratively incorporated, compared to its performance as a standalone model where only domain knowledge is leveraged. More generally, these findings suggest that different foundation models balance statistical fit and domain knowledge in distinct ways, leading to systematic differences in the causal graphs they generate.

Table 1: Experiment results showing how closely the DAGs generated by each model align with the ground truth

| Dataset | Method | SHD | Confidence |
|---------|--------|-----|------------|
| Real-world data: Amazon Supply Chain | PC Algorithm | 25 | |
| | LiNGAM | 22 | |
| | CausalFusion Claude 3.7 Sonnet | 6 | 0.80 |
| | CausalFusion Nova Pro | 3 | 0.80 |
| | CausalFusion Nova Premier | 17 | 1.00 |
| | First Iteration Claude 3.7 Sonnet | 6 | |
| | First Iteration Nova Pro | 6 | |
| | First Iteration Nova Premier | 11 | |
| | Ground Truth | 0 | |
| Synthetic data: Generic E-commerce | PC Algorithm | 20 | |
| | LiNGAM | 15 | |
| | CausalFusion Claude 3.7 Sonnet | 7 | 0.80 |
| | CausalFusion Nova Pro | 2 | 1.00 |
| | CausalFusion Nova Premier | 18 | 0.95 |
| | First Iteration Claude 3.7 Sonnet | 8 | |
| | First Iteration Nova Pro | 3 | |
| | First Iteration Nova Premier | 5 | |
| | Ground Truth | 0 | |

## 3.2 RQ2 - REPRODUCIBILITY

Beyond accuracy against ground truth, an essential property of causal discovery frameworks is their reproducibility: repeated runs under identical conditions should converge toward consistent DAGs. This aspect is particularly critical for methods involving LLMs, where stochastic decoding may lead to variability in outputs. To assess stability, we executed 20 independent runs for each foundational model and iteration of CausalFusion, recording the number of distinct DAGs generated across runs. The results are shown in Table 2. A highly stable configuration would yield only a single unique DAG, whereas larger counts indicate greater variability.

Table 2: Reproducibility analysis: average number of unique DAGs and reproducibility ratio across 20 runs by model (synthetic dataset). The reproducibility ratio is computed as $\frac{R-U}{R-1}$ where $R$ is the number of runs and $U$ the number of unique DAGs.

| Agent | Avg. Unique DAGs | Avg. Reproducibility Ratio | Std. Dev. |
|-------|------------------|----------------------------|-----------|
| Claude 3.7 Sonnet | 16.4 | 0.23 | 0.42 |
| Nova Premier | 2.0 | 0.95 | 0.05 |
| Nova Pro | 2.2 | 0.94 | 0.04 |

The results reveal notable differences in reproducibility across foundation models. Claude 3.7 Sonnet shows substantial variability, producing the full set of 20 distinct DAGs from iteration 2 onward, which indicates low stability under repeated sampling. By contrast, Nova Premier and Nova Pro are markedly more consistent, typically generating only one to three unique DAGs per iteration, with reproducibility ratios above 0.9. Taken together, these findings highlight that reproducibility is not guaranteed in LLM-based causal discovery and can vary systematically across models. From a methodological standpoint, this suggests that different foundation models embody distinct inductive mechanisms, which in turn affects the stability of the generated causal structures. More broadly the analysis underscores the importance of assessing not only accuracy (alignment with ground truth) but also reproducibility when integrating LLMs into causal discovery frameworks, since scientific utility depends on both dimensions.

## 3.3 RQ3 - REASONING DEPTH

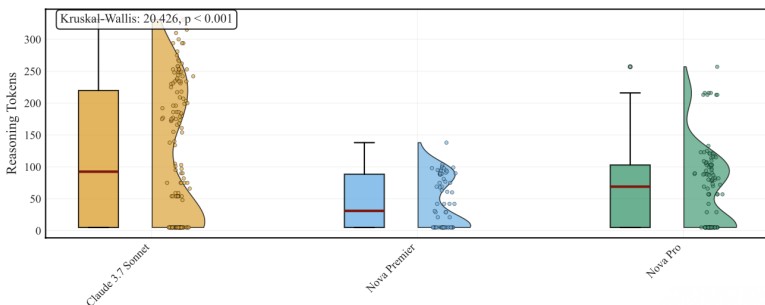

Figure 2: Distribution of reasoning tokens across FMs (synthetic dataset)

In addition, we assess SHD and reproducibility ratio against a proxy for chain-of-thought (CoT) depth to test whether deeper reasoning traces improve causal discovery performance. Given the absence of standardized metrics for CoT depth Wang et al. (2023), we operationalize depth as the number of reasoning tokens produced in the `ANALYSIS` field of each iteration. Outputs are tokenized using the `cl100k_base` tokenizer, and average token counts per model are reported. This enables us to investigate the hypothesis that increased reasoning depth correlates with improvements in DAG quality.

By comparing the performance of these models, we aim to test the broader hypothesis that increased CoT capacity correlates with improvements in causal discovery performance.

Table 3: Chain-of-thought tokens, performance metrics by model, and overall correlations.

| Model | CoT Tokens | SHD | Reproducibility Ratio |
|---|---|---|---|
| Claude 3.7 Sonnet | 118.4 | 6.5 | 0.23 |
| Nova Premier | 44.8 | 17.5 | 0.95 |
| Nova Pro | 64.9 | 2.5 | 0.94 |
| **Correlations across models** ($n = 3$) | | | |
| CoT vs SHD | | Pearson $r = -0.51$ | |
| CoT vs Reproducibility | | Pearson $r = -0.97$ | |

The descriptive analysis in Table 3 shows that the length of the analysis section, where the LLM justifies its refinement choices, varies across models. Correlation coefficients suggest that longer analyses are negatively associated with SHD ($r = -0.51$) and with reproducibility ($r = -0.97$), hinting that more extensive reasoning may relate to improved structural accuracy and stability. Given the very small sample size, these results should be seen as indicative rather than conclusive, and we note them here as a descriptive observation rather than a central finding.

## 4 CONCLUSION

Our experiments provide initial evidence that combining LLM reasoning with graph falsification feedback offers tangible advantages over traditional causal discovery methods. On synthetic data, CausalFusion achieved lower SHD than PC and LiNGAM, while on real-world supply chain data it again outperformed baseline approaches. Reproducibility analysis revealed notable differences across foundation models, underscoring model choice as a key factor for stability. Exploratory results further suggest that greater reasoning depth may contribute to both accuracy and consistency. These findings highlight the promise of LLM-assisted causal discovery while pointing to future research directions: enhancing reproducibility across models and extending evaluation to a broader range of datasets to assess generalizability.

## REPRODUCIBILITY STATEMENT

The code used to obtain experiments results in Table 1, 2, 3 is shared in a folder included in the current submission. The code corresponding to Table 1 also include the generation of the synthetic e-commerce dataset, along with all preprocessing steps and experiment configurations for both real world and synthetic experiment. For the real-world Amazon supply chain dataset, we provide detailed descriptions of the variables defined by domain experts, which form the basis of the ground-truth causal DAG used for benchmarking (see *Appendix B*). Preprocessing steps are also included in the code above. However, due to confidentiality constraints, the Amazon dataset cannot be shared at this stage. We are working toward obtaining approval to release this datasets ahead of a potential camera-ready submission. The LLM's knowledge base leveraged in the experiments, including causality principles, documentation and best practice, as well as dataset meta-data and general domain knowledge is also shared as part of the current submission. Furthermore, to ensure consistency, we fix random seeds in all experiments (seed = 42) and evaluate reproducibility explicitly by running each configuration 20 times, reporting variability across runs. We note that reproducibility has inherent limitations in the LLM setting, as stochastic decoding and differences across foundation models can lead to variability in outputs (see Section 2). Finally, hyperparameter settings are documented in Section 3 to further enhance reproducibility of both synthetic and real-world experiments.

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

APPENDIX A: CAUSALFUSION AGENT INTERACTION LOG

```
Iteration: 4
Model: Nova Pro
Agent ID: VWXI1K9IVO
Timestamp: 2025-09-12 18:39:12
============================================================
PROMPT:
============================================================
      ROLE: Expert Data Scientist specializing in COH causal analysis

      CURRENT EDGES: [('weather_tier', 'instation_backlog'),
      ('upstream_backlog', 'capped_out_hours'),
      ('daily_updated_cap_target', 'capped_out_hours'),
      ('latest_utilization', 'capped_out_hours'),
      ('latest_slammed_volume', 'capped_out_hours'),
      ('w1_capacity_ask', 'capped_out_hours')]

          GRAPH FALSIFICATION RESULTS:
          - Graph Informative: True
          - CI Violations %: 68.2%
          - CI Violations List: [(w1_capacity_ask ⊥
             ↪ daily_updated_cap_target | Pa(w1_capacity_ask)), ...]
          - DAG equivalent to randomly generated graphs?: p-value=0.050
          - Causal Minimality Violations List:
          [(capped_out_hours ⊥̸ upstream_backlog | Pa(capped_out_hours
             ↪ ) \ {upstream_backlog}), ...]

      KNOWLEDGE BOUNDARIES:
      When defining refined_edges, only include nodes that correspond
         ↪ to
      variables explicitly provided in your domain knowledge.
      Do not introduce or infer any additional variables beyond these
         ↪ features.

      FORBIDDEN: Creating new variables, abbreviations, or variations
      FORBIDDEN: Using any variable names not in the above list

      EXAMPLE OF REFINED EDGES:
      CORRECT: (weather_tier, instation_backlog)
      CORRECT: (upstream_backlog, capped_out_hours)
      WRONG: (weather, backlog) - abbreviated names not allowed
      WRONG: (productivity, queue_length) - new variables not allowed

      TASKS:
      1. Analyze test results using COH domain expertise
      2. Suggest refined or initial DAG edges based on causality
         ↪ principles. Only the AVAILABLE COH VARIABLES above are
         ↪ admitted as nodes.
      3. Provide confidence score (0.0-1.0) for current DAG
      4. Focus on reducing CI violations while maintaining domain
         ↪ validity

      OUTPUT FORMAT - Important: Always format your answers following
         ↪ this template:
      ANALYSIS: [Your expert analysis]
      REFINED EDGES: [List of (source, target) tuples]
      CONFIDENCE: [0.0-1.0]

      VALIDATION CHECK: Before finalizing, verify ALL variables in
         ↪ REFINED EDGES are from the allowed list above.

============================================================
AGENT RESPONSE:
============================================================
```

```
ANALYSIS: ANALYSIS: The current DAG shows a high percentage of CI
    ↪ violations (68.2%) and a potentially unnecessary edge between
    ↪ 'upstream_backlog' and 'capped_out_hours'. This suggests that the
    ↪ relationship between these two variables may not be as direct or
    ↪ strong as initially assumed. The random graph P-value of 0.050
    ↪ indicates that the current graph is significantly different from a
    ↪ random graph, although difference could be enhanced through
    ↪ further refinement.
As a result, the edge (upstream_backlog, capped_out_hours) has been
    ↪ removed due to its problematic nature as indicated by the graph
    ↪ falsification results. The remaining edges are maintained as they
    ↪ align with the domain knowledge and do not introduce new variables.

REFINED EDGES:
- (weather_tier, instation_backlog)
- (daily_updated_cap_target, capped_out_hours)
- (latest_utilization, capped_out_hours)
- (latest_slammed_volume, capped_out_hours)
- (w1_capacity_ask, capped_out_hours)

CONFIDENCE: 0.7
```

APPENDIX B: GROUND-TRUTH DAG FOR REAL-WORLD AMAZON SUPPLY CHAIN
EXPERIMENT: NODES DESCRIPTION

Table 4: Node descriptions and downstream effects in the causal DAG.

| Node | Description | Downstream Effects |
|------|-------------|--------------------|
| latest_slammed_volume | Slammed volume is considered a "source of truth" of what is already prepared to arrive at the delivery station (warehouse). | capped_out_hours |
| capped_out_hours | Capped Out Hours happen when a delivery station reaches its maximum capacity. This means the station cannot handle any more packages than what is already scheduled. This occurs when the package volume exceeds the station's capacity limit. We track how many hours a station stays in this "maxed out" state. | – |
| latest_utilization | Utilization shows how much of a station's available capacity is being used. It is calculated as slammed volume divided by station capacity. | capped_out_hours |
| DUCT | Daily Updated Caps Target (DUCT) is a capacity planning metric updated daily at 10:30 AM local time. It serves as a reference point for operations teams to understand expected capacity needs, compare against available capacity, and identify risks, helping leaders make informed allocation decisions. | capped_out_hours |
| instation_backlog | Refers to packages that have been received at the delivery station but not yet processed for delivery. These shipments are physically present and will undergo processing before reaching the customer. | capped_out_hours |
| weather_tier | Weather tiers are a classification system used to categorize the severity of weather events. | instation_backlog |
| upstream_backlog | Packages with estimated arrival dates less than or equal to the reporting date but that have not yet arrived at the delivery station. | instation_backlog |
| w1_capacity_ask | Represents the Week-minus-1 horizon capacity request, i.e., the expected volume at the station within the upcoming week. | capped_out_hours |

APPENDIX C: VISUALIZATION OF GROUND-TRUTH DAG FOR REAL-WORLD AMAZON SUPPLY CHAIN

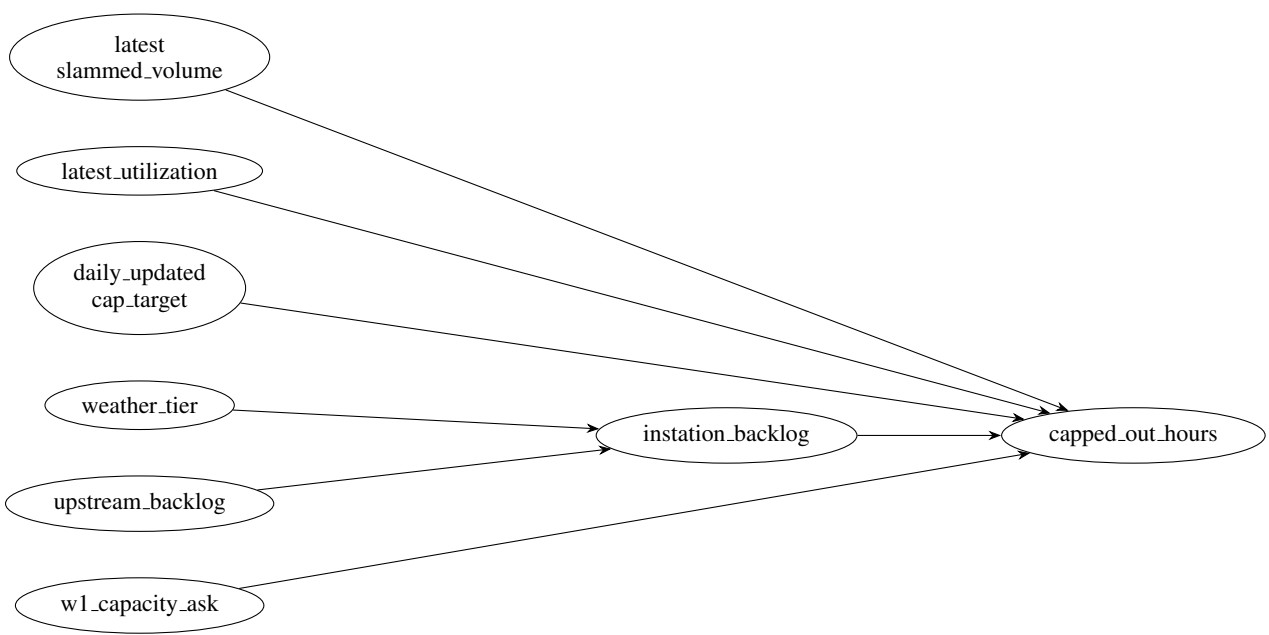

APPENDIX D: VISUALIZATION OF GROUND-TRUTH DAG FOR E-COMMERCE SYNTHETIC EXPERIMENT

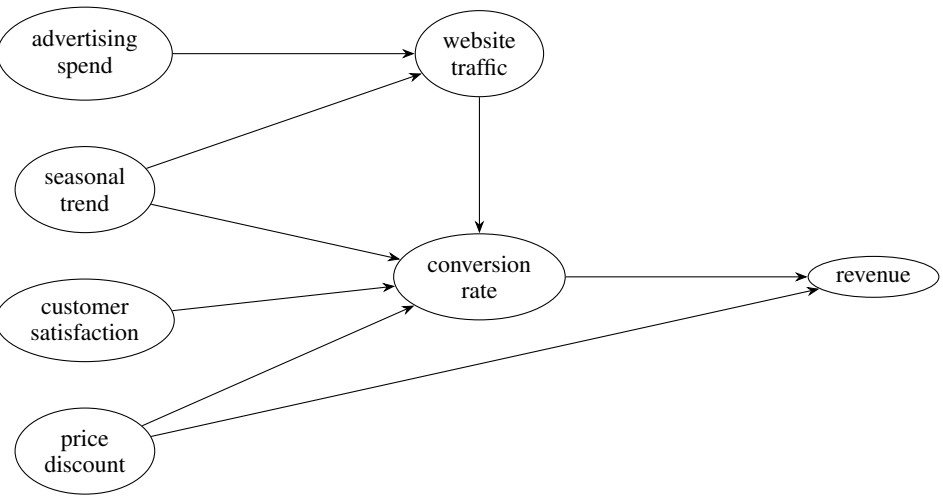