# OpenReview forum: "CausalFusion: Integrating LLMs and Graph Falsification for Causal Discovery"
_ICLR.cc/2026/Conference — Submitted to ICLR 2026_

### Official Review · Reviewer_2vBe · 2025-10-16

**Soundness:** 2
**Presentation:** 2
**Contribution:** 1
**Rating:** 2
**Confidence:** 5

**Summary:**

This paper proposes a framework called CausalFusion for causal discovery. CausalFusion integrates knowledge base, LLMs, and constraint based falsification procedure to develop a causal discovery algorithm. CausalFusion framework is evaluated on two small scale datasets using three LLMs as baselines.

**Strengths:**

1. The paper is easy to read and understand
2. The idea of integrating LLMs and Causality is interesting and has real-world applications.

**Weaknesses:**

Major:
1. Line 94 states that existing methods such as [1] use metadata alone for causal discovery. I believe this is incorrect: [1] uses LLM-generated causal knowledge to improve data-driven models such as the PC algorithm.

2. A significant amount of discussion and comparison with related literature is missing. For example, similar to the task considered in this paper, the references [1–5] below aim to integrate LLMs with data-driven causal discovery methods.

3. The underlying assumptions about the causal graph and the data-generating process — for example, absence of unobserved confounding — are not clearly articulated.

4. Experiments are performed on only two small-scale datasets. The performance improvements compared to baseline LLM models are not large.

5. The proposed method is not compared experimentally with the approaches in [1-5], which I believe already address the task at hand.

6. Although the Amazon dataset is claimed to be outside the pretraining data of pretrained LLMs, the variable names and the causal relationships among them are common in real-world applications; hence those causal relationships may be present in the LLMs’ training data. In the Results, the claim that “Nova Premier prioritized improving falsification outcomes far more strongly” highlights a limitation of the “applicability of falsification” module in the overall pipeline. The manuscript does not clearly articulate how to handle such scenarios.


Minor:
1. Line 42: Unlike traditional Shapley values, methods for causal Shapley values [3] exist to provide causal root-cause attributions.
2. Line 49: SCMs can be either parametric or nonparametric.
3. Figure 1 extends beyond the margins.
4. Line 214 — the hypothesis that chain-of-thought improves task performance, including for causal discovery, is already known.

References:

[1] Vashishtha, Aniket, et al. "Causal inference using LLM-guided discovery."

[2] Long, Stephanie, et al. "Causal discovery with language models as imperfect experts."

[3] Takayama, Masayuki, et al. "Integrating large language models in causal discovery: A statistical causal approach."

[4] Ban, Taiyu, et al. "From query tools to causal architects: Harnessing large language models for advanced causal discovery from data."

[5] Antonucci, Alessandro; Gregorio Piqué; Marco Zaffalon. "Zero-shot causal graph extrapolation from text via LLMs."

[6] Heskes, Tom, et al. "Causal Shapley values: Exploiting causal knowledge to explain individual predictions of complex models."

**Questions:**

Address the comments mentioned in weaknesses section.

---

### Official Review · Reviewer_8ek2 · 2025-11-01

**Soundness:** 3
**Presentation:** 3
**Contribution:** 2
**Rating:** 2
**Confidence:** 5

**Summary:**

** Summary

This work aims to construct a causal graph (DAG class) following the concept of constraint-based causal discovery methods based on hybrid data resources, namely, numerical data and background knowledge of LLMs. The pipeline starts with a graph constructed by LLMs, then, it iteratively refine the graph based on conditional independence tests based on numerical data and background knowledge/reasoning of LLMs to determine a DAG with the highest confidence.

** Recommendation

I would like to recommend a rejection for this paper for the limited novelty of the paper’s core idea and technique. As far as I know, several papers have proposed hybrid causal discovery methods based on numerical data and LLMs. However, this paper has its own merits (details provided later), though I think its contribution is not enough for ICLR.

**Strengths:**

1. The pipeline iteratively refines the causal graph, and provides the LLMs with reliable documents (e.g., the documentation of DoWhy) to help its reasoning process. This may potentially enhance LLMs’ understanding of the corresponding statistical principles.
2. The work uses relatively new benchmarks to evaluate the method. This is a convincing choice as many popular benchmarks are learned by LLMs (this is also mentioned in the paper).

**Weaknesses:**

1. My main concern is the novelty of the method as I mentioned in the recommendation part.
2. The reliability of the way that this work uses LLMs may need further clarification. For instance, it uses LLMs to generate a confidence score for candidate graphs. This may not be a convincing way of using LLMs, so it is better to do ablation study on this. Another example is that the pipeline starts with a LLM-based graph. If LLMs generate a graph that is far different from the truth, it might be hard to correct.
3. More experiments and descriptions may improve the paper quality. It can help understand more details of the tasks if the authors can add more details, e.g., the size of the benchmark dataset. It would be helpful if the authors can compare with more LLM-based/hybrid causal discovery methods.
4. It seems that the results obtained by the pipeline cannot significantly outperform those by simple querying LLMs.

**Questions:**

1. Have the authors tested whether providing the documentations (e.g., of DoWhy) significantly helps increase the performance or not?

---

### Official Review · Reviewer_9QpJ · 2025-11-02

**Soundness:** 2
**Presentation:** 3
**Contribution:** 2
**Rating:** 2
**Confidence:** 4

**Summary:**

CausalFusion introduces a hybrid causal discovery framework that combines LLMs acting as domain-expert agents with statistical graph falsification tests to iteratively refine causal DAGs. The LLM proposes candidate structures based on domain knowledge, while conditional independence, causal minimality, and falsifiability tests evaluate and provide feedback to improve them. This loop continues until confidence thresholds are met, producing graphs that balance empirical validity with expert plausibility. Experiments on synthetic and real-world (Amazon supply chain) data show that CausalFusion outperforms classical methods like PC and LiNGAM in structural accuracy, and that models with deeper reasoning traces generate more accurate and reproducible DAGs. Overall, it demonstrates a promising direction for fusing LLM reasoning with formal causal inference, though the analysis lacks empirical rigor needed to effectively test the framework's capability.

**Strengths:**

The paper has some strengths:

1. I really appreciate the authors focusing on new curated benchmarks with the help of domain experts to tackle the potential issue of data leakage. Prior literature focusing on LLMs for causal discovery fail to do this, and as recent papers have shown how the standard benchmarks are memorized, the proposed dataset can be an impactful contribution.

2. I appreciate the aim to develop a module which grounds LLMs in a principled approach. I also like adding dowhy's documentation for a structured framework.

3. The method mirrors the scientific process: the LLM hypothesizes a causal graph, empirical falsification tests evaluate it, and the LLM refines based on results enabling self-correcting causal discovery.

**Weaknesses:**

The work needs to add more empirical rigor:

1. SHD is a limited metric as shown by [1], authors should evaluate other metrics as well (like topological divergence, F1, Precision, Recall).

2. The authors need to evaluate and compare with prior works which also focus on integrating LLMs with traditional discovery algorithms like PC, CamML, etc. The authors have not covered the extensive literature which focuses on this issue, and does not do a convincing job of showing how their method stands out, or what does it have to contribute where prior hybrid methods fail. Ideally, the paper should compare with those methods also, instead of focusing on simple baselines like traditional methods, or LLM only baselines. Refer papers [1][2][3]

3. Utilizing LLMs to generate the confidence score is not a robust way of quantifying uncertainty. Many recent works show LLMs tend to be overconfident in their predictions [4][5]. Since the framework heavily relies on the confidence score generated by the model to requery until the final causal graph is achieved, this seems to be a flaw of the work.

4. The paper compares against PC and Lingam, but does not provide any description of the sample size used. Since PC tends to perform better with higher sample size, I am curious if the authors investigated that. Ideally the paper should include this detail as well.

5. The relationship between depth of CoT and discovery performance seems out of place.

6. Although reproducibility is one of their RQs, the results reveal high variability between runs for some models (e.g., Claude 3.7 Sonnet), yet the paper doesn’t analyze why the variance arises or propose methods to reduce it (e.g., temperature control, structured decoding)

7. All experiments are confined to one synthetic and one proprietary real-world dataset; there’s no test on diverse domains (e.g., healthcare, economics). Thus, it’s unclear whether the approach generalizes beyond narrowly defined settings.

The work seems to lack empirical rigor, and would benefit from expanding on its presented results by following the suggestions given above.

References:
[1] Causal Order: The Key to Leveraging Imperfect Experts in Causal Inference (Vashishtha et al.,2025)

[2] Efficient Causal Graph Discovery Using Large Language Models (Jiralerspong et al., 2024)

[3]  Causal Structure Learning Supervised by Large Language Model (Ban et al., 2023)

[4] Mind the Confidence Gap: Overconfidence, Calibration, and Distractor Effects in Large Language Models (Chhikara et al., 2025)

[5] Uncertainty Quantification and Confidence Calibration in Large Language Models: A Survey (Liu et al., 2025)

**Questions:**

Please refer the weaknesses section, and try to respond to each of the weaknesses, which I hope will help make your framework's generalization more evident. Provide more empirical support for your framework.

---

### Official Review · Reviewer_rBvi · 2025-11-06

**Soundness:** 2
**Presentation:** 1
**Contribution:** 1
**Rating:** 2
**Confidence:** 4

**Summary:**

This paper proposes a causal discovery method that incorporates both expert knowledge and available statistical information in its discovery process. It does this by using a language model to obtain an estimate of the underlying DAG which is then passed through a round of “falsification” to generate information on how faithful the estimated DAG is to  available statistical information. This information is then passed back to the LLM which incorporates it in its reasoning to provide a refined estimate of the DAG that is in greater alignment with available statistical information. The LLM is asked to produce (a) an updated DAG estimate, (b) an explanation for its revised estimate, and (c) a confidence score. The paper tests the method in two settings (a synthetic setting and one based on private real world data) to measure the accuracy of their method in comparison to statistical causal discovery methods and LLM based methods. It also studies the method’s reproducibility behaviour, i.e consistency in resultant DAG set, and the relation that reasoning depth has with accuracy and reproducibility.

**Strengths:**

To the best of my knowledge, “iterative falsification” appears to be a novel approach to combining LLM expert knowledge with available statistical information. Existing approaches often use LLM experts to generate prior estimates of the DAG which are then supplied to conventional causal discovery algorithms in some or the other way. Here the LLM takes a more active and central role in the discovery process.

**Weaknesses:**

Major Weaknesses (and related comments)

* The design of the synthetic experiment and motivation for specific design choices has been largely omitted (apart from the causal graph used). More information on the exact setup of the experiment, and more importantly  a justification for that setup should be included in the paper.
* A primary motivation for the proposed method was that expert-based causal discovery methods could not be scaled. However, there were no experiments in the paper that discussed the scaling behaviour of their methods, in fact no mention of this was made later.
* Experimental Results for RQ-1 show only marginal (if any) improvement at all. These relatively weak and limited experimental results are not very encouraging.
* Experiments for RQ2 are quite limited and do not provide a full picture on the matter of reproducibility in CausalFusion. Experiments with a greater number of agents, and perhaps strategies to improve low reproducibility (e.g. Claude 3.7 Sonnet) would greatly improve the quality of this section and the overall method itself.
* Results in RQ-3 do not add much to the discussion at all, computing correlation coefficients with only 3 samples does not make much sense. Moreover, the results and subsequent discussion do not provide a concrete answer to the question posed (RQ-3) in the first place.

Minor Weaknesses
* The paper is not well-written (e.g the paragraph at 069-079 and the paragraph at 080-092 are very similar with only minor differences). A variety of other minor writing issues makes it appear that not much effort was put into writing the paper.
* The method proposed could benefit from a change in terminology, is ‘falsification’ the most appropriate term for the iterative process described?

**Questions:**

Please see the weaknesses above.

Beyond them, could the authors clarify what exactly the random graph benchmark tests (2.3.3) are and how they provide relevant information to the LLM in the process of revising its DAG estimate?

---

### Meta-Review · Area_Chair_2wKS · 2026-01-04

**Summary:**

The reviewers' major concerns can be summarized as follows:

1. Limited Novelty: Multiple reviewers (8ek2, 2vBe) pointed out that the core idea of integrating domain knowledge (including that from LLMs) with data-driven causal discovery methods is not entirely new, as existing literature (such as the hybrid methods mentioned) has already explored this approach. The paper failed to clearly articulate the significant advantages or unique contributions of their method compared to these existing works.

2. Weak Experimental Evaluation:
- Limited Improvement in Results: Reviewer rBvi noted that the performance improvement demonstrated in the core experiments (RQ-1) was marginal at best, lacking persuasiveness.
- Over-reliance on a Single Metric: Reviewer 9QpJ argued that relying solely on the Structural Hamming Distance (SHD) is insufficient for comprehensively evaluating causal graphs and suggested incorporating more diverse metrics.
- Insufficient Baseline Comparisons: Reviewers 9QpJ and 2vBe both emphasized that the paper lacked thorough comparisons with existing advanced hybrid methods combining LLMs and causal discovery, instead primarily comparing against traditional algorithms like PC, LiNGAM, and LLM-only baselines. This makes it difficult to demonstrate the advancement of their method.
- Limited Scale and Generalizability: Reviewers 9QpJ and 2vBe believed that evaluation on only one synthetic dataset and one proprietary real-world dataset is insufficient to prove the method's generalizability. The lack of experimental details (such as sample size) also undermined the credibility of the results.

3. Methodological Reliability Issues:
- Reviewer 9QpJ expressed concern about relying on LLM-generated confidence scores, noting that LLMs are prone to overconfidence, which could compromise the reliability of the iterative refinement process.
- Reviewers 8ek2 and 2vBe were concerned about the quality of the initial graph generated by the LLM. They worried that if the initial graph is significantly biased, subsequent corrections might struggle to converge to the correct structure.

4. Insufficient Analysis and Writing Depth:
- Reviewer rBvi considered the writing unpolished, with repetitive paragraphs and insufficient explanation for the motivation behind experimental design choices.
- Reviewer rBvi also indicated that the analysis of the relationship between the depth of chain-of-thought and performance (RQ-3) was overly simplistic (based on correlation calculations with only 3 samples), unconvincing, and the reproducibility analysis (RQ-2) was also inadequate.

**Reviewer Concerns:**

Since the authors did not submit a rebuttal, all concerns raised by the reviewers remained unaddressed.

**Reviewer Scores:**

Given that the authors did not participate in the rebuttal discussion, it is reasonable to conclude that the reviewers' scores would likely have remained stable or even decreased slightly after seeing other reviews, as their pointed-out core issues were consensus-based and went unanswered.
- Reviewer rBvi: Original score: 2. After seeing other reviewers also highlighting issues with novelty, experimental rigor, and analytical depth, this reviewer's score would highly likely remain unchanged.
- Reviewer 9QpJ: Original score: 2. The weaknesses raised by this reviewer regarding experimental rigor and lack of method comparisons were echoed by others, potentially making them adhere more firmly to the original score.
- Reviewer 8ek2: Original score: 2. The direct challenge to novelty was this reviewer's central stance, and with support from other reviewers' opinions, changing the score is unlikely.
- Reviewer 2vBe: Original score: 2. The concerns about missing relevant literature and insufficient experimental comparisons raised by this reviewer were similarly left unresolved.

---

### Decision · Program_Chairs · 2026-01-26

Reject